# Using Surface Topography to Visualize Spinal Motion During Gait—Examples of Possible Applications and All Tools for Open Science

**DOI:** 10.3390/bioengineering12040348

**Published:** 2025-03-28

**Authors:** Jürgen Konradi, Ulrich Betz, Janine Huthwelker, Claudia Wolf, Irene Schmidtmann, Ruben Westphal, Meghan Cerpa, Lawrence G. Lenke, Philipp Drees

**Affiliations:** 1Institute of Physical Therapy, Prevention and Rehabilitation, University Medical Center of the Johannes Gutenberg University Mainz, Langenbeckstrasse 1, D-55131 Mainz, Germany; ulrich.betz@unimedizin-mainz.de (U.B.); janine.huthwelker@unimedizin-mainz.de (J.H.); claudia.wolf@unimedizin-mainz.de (C.W.); 2Institute of Medical Biostatistics, Epidemiology and Informatics, University Medical Center of the Johannes Gutenberg University Mainz, Rhabanusstraße 3/Tower A, D-55118 Mainz, Germany; irene.schmidtmann@unimedizin-mainz.de (I.S.); rubenswestphal@gmail.com (R.W.); 3Department of Orthopedic Surgery, Columbia University Medical Center, 5141 Broadway, New York, NY 10032, USA; meghancerpa@yahoo.com (M.C.); ll2989@cumc.columbia.edu (L.G.L.); 4Department of Orthopedics and Trauma Surgery, University Medical Center of the Johannes Gutenberg University Mainz, Langenbeckstrasse 1, D-55131 Mainz, Germany; philipp.drees@unimedizin-mainz.de

**Keywords:** spine biomechanics, graph-based representation, motion analysis, surface topography, rasterstereography

## Abstract

Precise segmental spinal analysis during gait has various implications for clinical use and basic research. Here, we report the use of Surface Topography (ST) to analyze three-dimensional spinal segment movements, in combination with foot pressure measuring, to describe individual vertebral bodies’ motion relative to specific phases of gait. Using Statistical Analysis System (SAS) scripts, single files were merged into one raw data table and were used to generate a standardized gait cycle (SGC) for each measurement, including all measured gait cycles for each individual patient, with a spline function to obtain smooth curve progressions. Graph templates from Statistical Package for the Social Sciences create detailed visualizations of the SGCs. Previously obtained measurements from healthy participants were used to demonstrate possible applications of our method. An impressive inter-individual variability as well as intra-individual consistency of spinal motion is shown. The transformation into an SGC facilitates intra- and inter-individual comparisons for qualitative and quantitative analyses. In future studies, we want to use this method to distinguish between physiologic and pathologic spinal motion. Artificial intelligence-based analysis can facilitate this process. All tools and visualizations used are freely available in repositories to enable the replication and validation of our findings.

## 1. Introduction

Precise segmental spinal analysis during gait would have various implications for basic research and clinical use, exemplarily in the context of low back pain (LBP). LBP in the German population has a point prevalence of 25–40%, a 12-month prevalence of 60–70% [1], and a lifetime prevalence in the American population of up to 85% [2] in adults. According to the German Medical Association [3], LBP is currently ranked as the most frequent musculoskeletal disorder, with an annual cost of 3.6 billion Euros. Thus, companies and stakeholders have a strong interest in reducing such often work-related diseases [4]. As many as 90% of LBP complaints have no anatomic structure abnormalities that can be identified as the source of the patient’s pathologies [5]. Most of these occur in motion [6]. Hence, static and structure-orientated diagnostic approaches like X-rays or MRI cannot detect the etiology of pathology in the majority of LBP patients, meanwhile incurring unnecessary costs. Therefore, systems for multidimensional motion analysis are becoming instrumental in the diagnosis of unspecific musculoskeletal problems, as they are able to provide additional dynamic and functional information for individualized diagnoses [7], even though they do not provide ground-truth imaging, such as fluoroscopy or dynamic X-ray imaging.

Even though this three-dimensional approach is popular for musculoskeletal problems of the pelvic–leg region, the spine and trunk are often neglected due to metrological limitations [8,9]. For example, the assessment of each functional spinal unit requires the application of three non-collinear markers per segment [9]. Due to the close anatomical vicinity of adjacent vertebrae, unintended marker contact can cause significant measuring artifacts. Furthermore, due to the variety of spinal segments, a complex preparation is required, which is immensely prone to palpation bias and can result in measurement error [8]. Even though current research recognizes the spine’s active role in balance and locomotion dynamics [10], truncal measurements in instrumented gait analysis mostly regard it as a rigid body [11], usually called the “passenger unit”, which is transported by the “locomotor”, with no relevant contribution to ambulation [12]. Based on the described limitations, there is little literature regarding three-dimensional segment-related spinal movement during gait. Additionally, the results of the few existing reports are not comparable because of methodological differences, ranging from three-dimensional motion analyses of isolated spinal segments to invasive measurement procedures [13,14,15,16,17,18,19,20,21,22,23].

The most valid method [19] uses markers to capture three-dimensional motion by inserting bone pins under local anesthesia into the spinal processes of each lumbar vertebra under the control of an image converter. This procedure can theoretically be considered a gold standard [24]. However, due to the surgical invasiveness and the resulting radiation exposure, this approach is inappropriate for use as a routine assessment in clinical as well as in scientific practice. Furthermore, only very low amplitudes of spinal motion have been observed, which may be either influenced by pain-induced inhibition of habitual movement or by residual effects of local anesthesia.

Hence, the ability to measure the spinal motion of each single segment during gait without extensive preparation or the usage of invasive or radiation-based measurement approaches, however, is a valuable tool for clinical practice as well as basic research. It can expand on our knowledge of the spine’s role in maintaining balance and upright posture during gait, as well as provide further understanding of the underlying biomechanical mechanisms behind unspecific musculoskeletal conditions, such as LBP [24]. Rasterstereography (RS), more recently called Surface Topography (ST) [25], is a non-invasive and reliable [26] alternative high-precision technique to analyze the shapes of surfaces [27], even in 360° [28]. Resting upon back shape data and orientated on visible anatomical landmarks, the Turner–Smith model [29] combined with other models [30,31,32,33] can be used for the estimation of the segmental spine posture. Originally, it was used for static or quasi-dynamic measurements during stance [25], specifically within the context of scoliosis [34,35,36,37]. However, its use in the setting of degenerative disk disease has been questioned [38]. More recently, dynamic measurements have been introduced [39]. Using DIERS formetric’s standard software during gait analysis, its reliability to detect certain measuring points (e.g., max of T4) at some point during the gait cycle has been demonstrated (Gipsman et al., 2014) [40]. What former ST approaches lacked were the precise determination of spinal movement in direct relation to phases of gait during the gait cycle, and subsequent standardization of data from various gait cycles to make data intra- and inter-individually comparable, regardless of aspects confounded by walking speed or stride length. As already demonstrated [41,42], we successfully further developed this method in this direction.

In our approach, we utilized DIERS formetric as a means for gathering dynamic ST measurements. The system generates 3D images of the surface, estimates corresponding 3D movements of the spine for each individual segment starting at vertebra prominens [30] and ending at the pelvis [31], and features a treadmill with an embedded foot pressure measuring plate to analyze ground reaction forces. This can be used to identify certain moments in gait, as gait follows certain identifiable determinants [43]. According to the common model [12], a gait cycle can be divided into two periods (60% stance and 40% swing) and consists of eight total phases. The most relevant phases pertaining to this study are Initial Contact (IC), which divides gait cycles, and Initial Swing (IS), which departs the stance from the swing phase.

In this methodologic contribution, we aim to fully visualize and describe spinal movement in direct relation to gait phases, thereby concentrating on spinal rotation. We therefore used previously obtained measurements from healthy participants. First, we describe the use of foot pressure measuring data to encode for the step and swing phases as well as for complete gait cycles. Secondly, there had to be a modification of the system’s export functions in order to synchronize spinal motion data with foot pressure measuring data and combine them into the same raw data export. Since all exports are separate for each measurement, the third task was to merge single export files to create a complete raw data table. Finally, we were able to then standardize the combined raw data set of three or more gait cycles by interpolating splines to make spinal motion analysis relative to gait cycles intra- and inter-individually comparable. Together, this enables us to create oscillographs of spinal movement for and across each gait cycle, resulting in interpretable depths and precision for analyses. This analysis will address the described methods separately and provide the developed solutions for all spinal movement oscillographs of individual gait patterns as well as their standardized counterparts in several repositories [44,45,46,47,48,49,50].

## 2. Materials and Methods

In this methodologic contribution, we use previously obtained dynamic ST measurements from 201 healthy participants (132 females, 69 males, aged 18 to 70) that have already been used as a normative reference data pool. All of them had passed several functional tests and very strict inclusion criteria [41]. They were taken with a 4D spine and posture analysis device (DIERS Formetric III 4D™, DICAM v3.7Beta, Wiesbaden, Germany). It projects structured light onto the textile-free back of the individual person. A camera unit in defined positions records the movement with a frequency of 60 Hz. The software analyzes all three dimensions of each individual measuring point (up to 150,000, depending on body size) and generates a 3D image of the surface. The system then calculates [31] the corresponding 3D movements for each spinal segment, starting at the spinous process of C7 and ending at the pelvis. Due to the low correlation accuracy between surface structure and spinal position in the lower back area, L5 is not measured. Additionally, it is supplemented by one rear axis and two lateral cameras that record a video signal, enabling a subsequent visual inspection for multiple purposes. The measurement setup can be seen in Figure 1.

An integrated Zebris™ foot pressure measuring plate (5376 sensors, scanning frequency 120 Hz, sensitivity 1 N/cm^2^, accuracy 5% FS) enables the analysis of ground reaction forces.

When beginning the analysis, all measuring devices start simultaneously, but due to the different measuring frequencies this results in an unequal number of observation times. This first needs to be reconciled in order to enable analysis of spinal motion directly related to gait.

**Figure 1 bioengineering-12-00348-f001:**
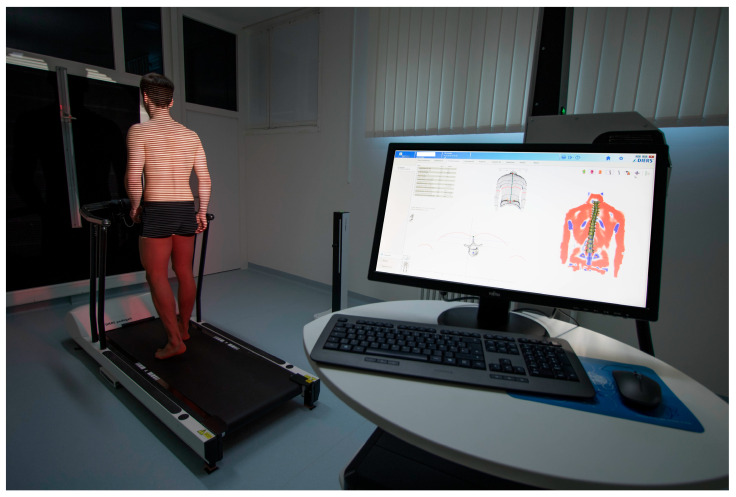
Dynamic Surface Topography measuring setup. Participant walking on a treadmill with an integrated foot pressure measuring plate while structured light is projected on the textile-free back. System generates a 3D image of the surface and then calculates the corresponding 3D movements for each spinal segment.

Therefore, the necessary data were obtained from an already existing normative referent data pool [41]. The study was approved by the responsible ethics committee of the medical chamber Rhineland-Palatinate (837.194.16) and is registered with WHO (INT: DRKS00010834). All participants gave informed consent. For further information on the ST measuring device, experimental procedures, or characteristics of participants, refer to Huthwelker et al. (2022) [41].

Here, we provide detailed descriptions of the four central processes.

### 2.1. Encoding of Step and Swing Phases and Complete Gait Cycles into the Spinal Model

In our approach, we started to measure a gait cycle with the first full IC of the right foot, which is also the start of encoding for the stance phase right. Hence, the start of the next gait cycle is the next IC of the right foot and so forth. Both leg swing phases are also marked in the data export. Along their assigned time stamp, the respective gait phases were encoded and synchronized into the raw data of the spinal model. We introduced the notion to the manufacturer DIERS. They implemented the concept in the software (DICAM v3.7Beta). We validated the updated software by taking a random sample of 20 participants’ video recordings of the back and lateral axis cameras in which single frames were checked for face-valid results compared to the automated detection. In all recordings, the visually identified moment in time of IC was in the range of ±3 Frames/3/60 Hz compared to the frame number in the exported raw data. For a detailed explanation of the notion, compare the related repository [46]. Since all observed video recordings revealed valid results, the next data step could be addressed.

### 2.2. Assembly of Individual Measurement Export Files and Creation of Rotation Graphs

DICAM evaluates over one hundred global (e.g., lordosis and kyphosis angles) and local (e.g., 3D position data of each vertebra) parameters of the spine that can be exported as raw data in .CSV format. The exported data are separate for each measurement that was chosen in the menu. Depending on the parameter choice, different columns are generated. Based on the parameter settings, specific column distribution results are produced. A Statistical Analysis System (SAS v9.4) syntax script assembles all single exports into one complete raw data table [44]. Since not all characters in the DICAM export are compatible with SAS, the script defines its own column headings and labels them accordingly. Consequently, the parameter settings of DICAM are crucial; otherwise, wrong data would be read in. Columns containing the subject’s code and the speed in km/h are imported based on the filename of the former DICAM export. Now, all kinds of statistical analyses can be applied to a full set of data.

The resulting complete raw data table can also be imported by Statistical Package for the Social Sciences (SPSS v23). We used this to check data for plausibility and potential outlying values and to generate oscillation graphs based on individual movement data with a direct relation to the phases of gait. For the graphs, we used the position data of the vertebra and the pelvis in the transversal plane. For the indication of gait cycles, we used foot pressure measuring data indicating the phases of gait. A graph template, as well as the respective SPSS script for execution, is openly provided [49].

### 2.3. Standardization Combining Raw Data of Three or More Gait Cycles for Interpolating Splines and Creation of Rotational Graphs

We used an openly available SAS (v9.4) syntax script [44] to generate a standardized gait cycle (SGC) for each measurement. By computing a standardized gait cycle on a scale of 0–100%, measurements all differing in the number of observations are thus made comparable. One SGC per measurement was generated out of three (all available) gait cycles by using a spline function as part of the syntax. The resulting raw data table was analyzed using SPSS to create rotational graphs of the spine now within an SGC. Therefore, another graph template and the respective executing SPSS script were created and made openly available [48]. It can be downloaded and used on its own measurement data from a similar ST device after files have been exported.

## 3. Results

After visual inspection of all spinal rotation graphs of our 201 participants, we present a selection of interesting cases and related motion analytical considerations in the following subsection. The purpose is to provide an overview of typical findings and emerging patterns as well as highlight rare cases. Each of these graphs (Figure 2, Figure 3, Figure 4 and Figure 5) represents the spinal motion of one of our participants across three gait cycles and the same subject again (Figure 6, Figure 7, Figure 8 and Figure 9) after computing an SGC. Therefore, we analyzed average walking speeds of approximately 82–84 m per minute/5 km/h [12] as this speed provides data for most habitual movement patterns. In addition, we limited visualizations of rotational curves in the transverse plane as a first approach to make spinal motion visible. All anonymized single graphs containing only the graph number, but not the subject code, and an all-encompassing visualization are openly available [50].

### 3.1. Raw Data Visualizations of Rotational Curves Directly Related to Phases Gait

The pelvis and the lumbar segments show an opposite progression in comparison to the thoracic segments. The rotational direction changes gradually through each of the segments. As expected, periodic near sinusoidal oscillations are seen, revealing a phase shift between the pelvis and the upper thoracic segments with their maxima facing each other, meaning a direct equalization of the pelvic rotation by the thoracic spine (Figure 2). At IC right, the pelvis and L4 are maximally rotated to the left, reaching the zero point in the middle of the stance phase. L4 constantly follows the rotation of the pelvis. T12 has a fairly small amplitude, being close to the zero point where the intersection of movement directions takes place. Maximum antagonistic rotation is displayed by T7 and T8. The movement direction then changes again, with T4 also rotating in the opposite direction of the pelvis but to a lesser extent. Along the gait cycle, the amplitude and the period are intra-individually constant. Even though we see expected movement patterns, there are also broad variations with specific manifestations for each individual (Figure 3, Figure 4 and Figure 5).

**Figure 2 bioengineering-12-00348-f002:**
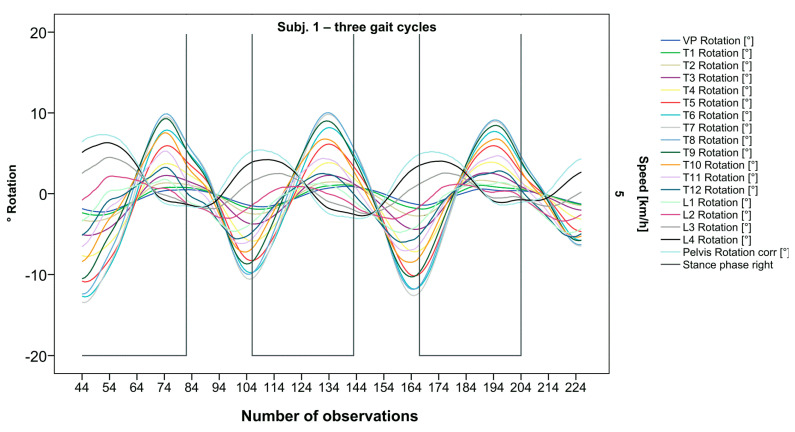
Visual representation for all segments of graph Nr.641 [50]. Positive values show rotation to the left, and negative values show rotation to the right. Observation number is displayed on the abscissa, always starting with Initial Contact of the right foot. Durations of right stance phases are delineated with a vertical black line.

**Figure 3 bioengineering-12-00348-f003:**
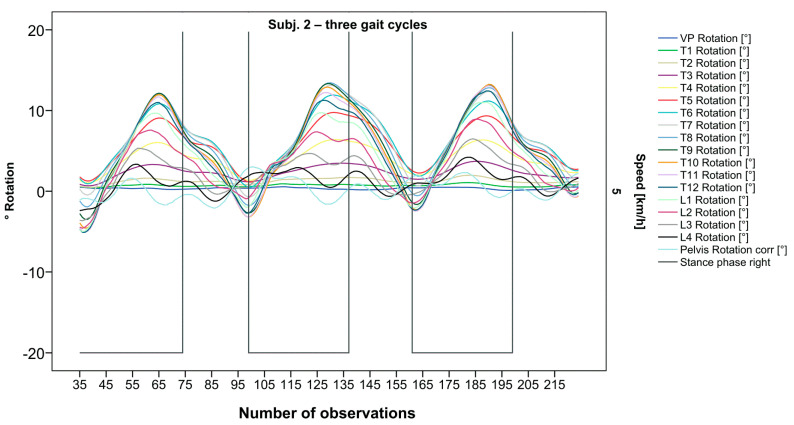
Illustration of all segments of graph Nr.622 [50]. Beginning at the pelvis, rhythmic movements superimpose the curve progressions of all vertebral bodies upward. Durations of right stance phases are delineated with a vertical black line.

**Figure 4 bioengineering-12-00348-f004:**
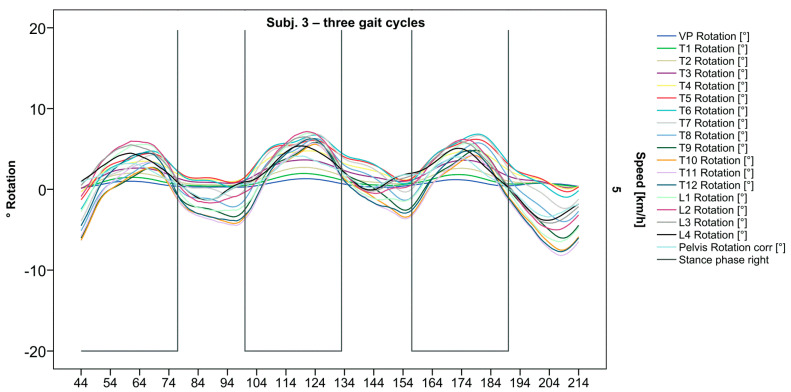
Visual representation of graph Nr.615 [50] depicting a rarely seen reverse pattern. Most parts of the spine are rotating nearly in phase. Durations of right stance phases are delineated with a vertical black line.

**Figure 5 bioengineering-12-00348-f005:**
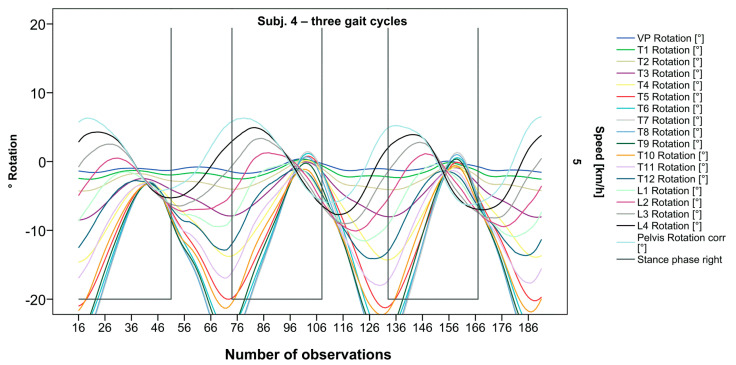
Visual representation of graph Nr.608 [50] depicting a shift of rotation to the right for the entire thoracic spine. Durations of right stance phases are delineated with a vertical black line.

After visual analysis of all individual cases, we made the following observations: Graphs of rotation patterns are usually characterized by sinusoidal curve progression (Figure 2). Rhythmic movements can superimpose these typical curves (Figure 3). They can be assumed as individually characterizing features and not attributed to measuring artifacts as they appear consistently throughout the whole spine and across all gait cycles. In particular cases, potentially non-sinusoidal but still periodic curves, mostly with steep rises, occur (Figure 3). Rarely, for instance, in the pelvis (Figure 3), very little movement (<5°) or even no systematic curve course could be detected. The measured values of individual segments oscillate around a “stable level” but not necessarily around the zero point. Frequently, this “symmetry line” (SL) of oscillation is shifted several degrees into one of the two directions of movement (Figure 5); this “level shift” (LS) might depend on the alignment during stance. During gait cycles, the amplitude and the period are intra-individually constant (Figure 2, Figure 3, Figure 4 and Figure 5) and behave relative to the stance phase for all participants but in different ways, varying for each individual. The direction of movement of the pelvis and the lower lumbar spine is usually opposite to that of the upper lumbar and thoracic spine. However, in some cases, the pelvis and the majority of all segments rotate in the same phase (Figure 4). Usually, the graphs of neighboring vertebral bodies rotate nearly in phase but with differently prominent maximum and minimum segmental motions. Maximum antagonistic rotation to the pelvis is mostly displayed by T8 (Figure 2). The movement excursions of these two regions can be equally large, or they can differ significantly. The “point of intersection” (PoI), the height of the segment where the two directions of movement exchange, varies depending on the subject and can be between the middle lumbar and lower thoracic spine. For the resulting phase shifts between these “counterparts”, we found multiple patterns reaching from exact antagonistic 100% (180°, sine-to-sine), over 50% (90°, sine-to-cosine) to 0% (0°, oscillating in phase). Relative to the stance phase of the right leg, the rotational maximum of the pelvis to the left predominately occurs between IC and mid stance.

In order to make the described spinal motion analysis intra- and inter-individually comparable, we had to standardize combined raw data of all gait cycles for interpolating splines.

### 3.2. Standardization of Combining Raw Data of Three or More Gait Cycles for Interpolating Splines and Creation of Rotational Graphs

After standardization, graphs combining raw data of all gait cycles can be created. In order to demonstrate the effects of standardization and interpolation, subsequent figures take up their counterparts from the previous section. Comparing Figure 2 and Figure 6, vertebral bodies of the middle and lower thoracic spine during the first and third gait cycle, previously presenting an asymmetrical curve progression, especially toward the left (Figure 2), now show a symmetrical curve progression leading to a much more precise identification of maxima (Figure 6). In given contexts, where rhythmic movements superimpose curve progressions, thereby constituting individually characterizing features (Figure 3), the standardization, nevertheless, preserves them, meanwhile improving maxima identification (Figure 7). The standardization not only enables comparability but can also clarify individual features. At first, Figure 4 reveals that the pelvis, lumbar spine, and all thoracic segments are rotating nearly in phase, but after transformation in SGC, we see that this must be subdivided so that the lumbar and middle thoracic spine rotate nearly in phase while the upper thoracic spine displays hardly any movement (Figure 8). A similar clarification of an individual feature occurs when comparing the LS to the right of the SL before (Figure 5) and after standardization (Figure 9), when the isolated LS of the thoracic spine and T12, being the PoI, is much easier to recognize. All single graphs within an SGC and an all-encompassing video are openly available [47].

**Figure 6 bioengineering-12-00348-f006:**
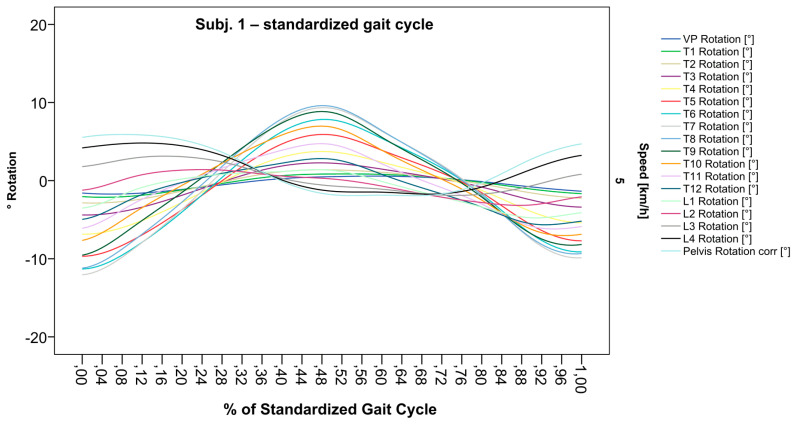
Illustration of all segments of graph Nr.641 SGC [47]. Near sinusoidal wave form, a further specified identification of maxima (rotation to the left) for all vertebral bodies occurred.

**Figure 7 bioengineering-12-00348-f007:**
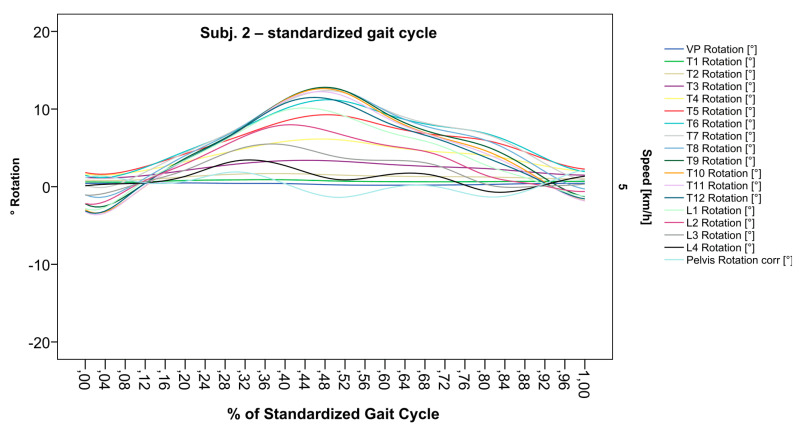
Illustration of all segments of graph Nr.622 SGC [47]. Superimposed oscillation as an individually characteristic feature is still visible; maxima identification nevertheless much more precise.

**Figure 8 bioengineering-12-00348-f008:**
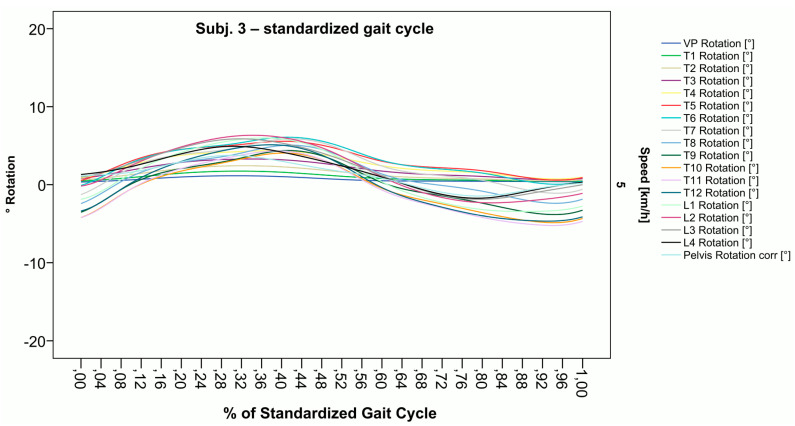
Illustration of all segments of graph Nr.613 SGC [47]. Isolated rotation in the phase of the lumbar and middle thoracic spine, as the individual feature becomes more apparent after transformation in SGC.

**Figure 9 bioengineering-12-00348-f009:**
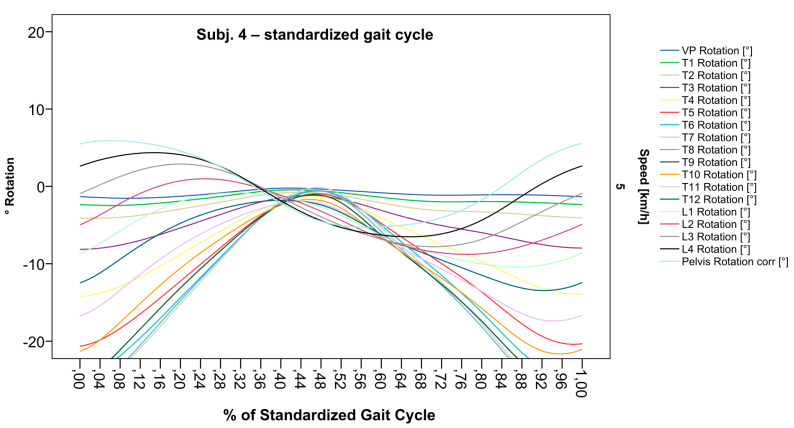
Illustration of all segments of graph Nr.608 SGC [47]. The isolated thoracic rotation shift to the right, with T12 being the point of intersection, is now easier to recognize.

## 4. Discussion

With the described methodologic advancements, ST can be used to visualize spinal motion as it directly relates to phases of gait and, after standardization, to compare these results intra- and inter-individually. As demonstrated, interpolating spline functions work for average walking speed measurements, leading to a more precise determination of relevant and characteristic points (e.g., maxima, phase shifts, lumbar and thoracic movement behavior), thereby aiding in the clarification of individual features. Additionally, this constitutes the basis for calculating phase shifts between different vertebral bodies as a future parameter to describe patterns of spinal motion in gait.

Using this method, we observed high intra-individual consistency of movement patterns as well as extensive inter-individual variation. When applying AI-based analysis, e.g., a Siamese neural network architecture, we were already able to identify 100% of individuals, thus constituting a spinal fingerprint [51]. This individuality of motion patterns is similar to previous work [17,19]. What contradicts these former findings is that our results detected regions (T6–T8) of many volunteers where they actually contained the largest movement excursions [42], whereas previous work detected the least spinal motion. Especially in regards to phase shift patterns of spinal segments, we detected subjects where the majority of all segments rotated in the same phase. One would expect this phenomenon while walking in amble. In this unusual pattern for humans, the ipsilateral instead of the contralateral arm is advanced by the leg. This type of ambulation is normally restricted to quadruped mammals, such as the elephant, but can also be examined by primate species [52], although this finding requires further examination. Taken together, our approach can directly relate segmental data to specific phases of gait and moments in time for an SGC.

Regarding the interpretation of normative reference data of asymptomatic healthy controls [42], the inter-individual variation due to differences in gait types, the alignment during stance [41,53], and other confounding factors must be determined. It is still unclear to which extent this would help to discern between physiological and pathological movement patterns. Thus, a relevant goal of future research will be to identify movement parameters and resulting characteristic patterns in which Artificial Intelligence-based analysis could be very helpful, as has been demonstrated for gait patterns [54]. As an example of dynamic ST, longitudinal changes of similarity in subjects might be of interest for the detection of early change, thereby supporting experts with an objective metric that might be helpful when approaching pathologies [51]. In order to test these assumptions, we will look for differences in spinal motion between the healthy controls and back pain patients, as well as other musculoskeletal disorders, in separate projects.

Limitations arise from the methodology, as reliability, reproducibility, and intra-individual consistency for the use in dynamic gait analysis have been shown [40,51], though validity is only for dynamic stance measures [55]. Although the transfer from validated stance measurements to dynamic gait analysis is feasible, there has not been validation of the spinal model in dynamic ST measurements so far. Hence, dynamic ST represents 3D spinal position data estimated from the back surface, which can be affected by contraction of the back muscles or movements and deformations of the soft tissue. Therefore, we were only able to apply internal validity measures to our results, e.g., the slightly displaced but still parallel course of the pelvis and L4. Furthermore, we only measured three gait cycles, and we need to investigate whether the inclusion of more than three gait cycles would alter the reported results. Alternatively, local Fourier transformations could also serve the same purpose in a superior way. Further research is needed to identify the most appropriate method of gait analysis necessary for adequate assessments of slower speeds in the setting of patients with back pain or hip/knee arthrosis where pathology inhibits walking (reducing speed).

Comprehension of these relationships will facilitate future research to understand the nature of pathologies, for example, back pain, arthrosis, scoliosis, and the effect of orthopedic surgery on spinal motion for comparison between physiological and pathological variations.

## 5. Conclusions

As the result of our methodologic advancement, ST now enables a face-valid and reproducible description of estimated spinal motion in direct relation to gait without extensive preparation procedures, significant radiation exposure, or other forms of invasive strategies. The transformation into an SGC facilitates intra- and inter-individual comparisons while preserving individual characteristic features. Hence, we conclude that this novel form of gait-related spinal motion analysis appears to have several advantages over existing methodology and holds much promise for future research in this field.

Regarding necessary future work and in order to address the lack of validity for dynamic ST, we propose two approaches: As a direct validation of dynamic ST, when compared to a gold standard, we suggest the application of 3D X-ray while walking on a treadmill (e.g., EOS). As a clinical challenge for dynamic ST, in order to provide even more face validity, a research project could evaluate the potential of ST to detect spinal fusions.

Nevertheless, before dynamic ST has been validated with a gold-standard measurement, crucial clinical decisions should not solely rely on its results.

## Data Availability

To enable our own studies and for the replication of our findings, all developed tools are provided in several publicly available repositories that can be found in the references [41,42,43,44,45,46,47]. This includes the aggregated data for the oscillographs. The underlying participant data could not be provided since the original informed consent only enables sharing with one research partner. Furthermore, public accessibility is not covered by the given ethics vote.

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
