# Peer review of "Using Surface Topography to Visualize Spinal Motion During Gait—Examples of Possible Applications and All Tools for Open Science"

_bioengineering, 2025, doi:10.3390/bioengineering12040348_

Round 1
Reviewer 1 Report
Comments and Suggestions for Authors
It is unclear how segmental spinal analysis could be used to improve basic research, as it is mentioned multiple times without clear indications or explanations of its practical applications.
More details about the results are missing in the abstract, particularly regarding how "AI-based analysis can facilitate this process."
The introduction does not mention alternative approaches for ground-truth imaging, such as fluoroscopy or dynamic X-ray imaging, which should be addressed for a more comprehensive overview.
The acquisition protocol is difficult to follow without an adequate figure to illustrate the process.
Key participant details such as age, BMI, height/weight, and other relevant characteristics are missing from the Materials and Methods section.
Although visualizations are provided, the accompanying narrative could be simplified to better highlight findings. Currently, it is challenging to follow the lines on the graphs (consider using distinct line types or markers to differentiate lumbar and thoracic vertebrae). Including a summary table of key results (e.g., phase shifts, amplitude ranges) would enhance reader comprehension and provide a more accessible reference.
The discussion occasionally references prior studies but does not critically engage with them. Furthermore, some of the studies cited appear outdated and should be reconsidered.
SPSS software was used for statistical analysis, but the results lack numerical support to substantiate the significant claim in the conclusion that "our concept enables precise description of spinal motion."
While AI is briefly mentioned, outlining more specific plans or methodologies for incorporating it into spinal motion analysis would significantly strengthen the conclusion.
Comments on the Quality of English LanguageThe English could be improved to more clearly express the research.
There are typos.
The article uses technical jargon and dense sentences that may hinder understanding for non-specialists.
Reviewer 2 Report
Comments and Suggestions for Authors
The current study aimed to visualize and describe spinal rotation in direct relation to gait phases by focusing on several key objectives. First, it encodes step and swing phases as well as complete gait cycles using foot pressure measurement data. To achieve this, the system’s export functions were modified to synchronize spinal motion data with foot pressure data in a single raw data export. Subsequently, individual export files were merged into a comprehensive raw data table. The combined dataset was then standardized across multiple gait cycles using spline interpolation, facilitating intra- and inter-individual comparisons. These advancements enable the creation of detailed oscillographs of spinal movement for both individual and standardized gait cycles, providing precision and interpretability for further analysis.
The study is of interest, but several parts need more information and rearrangements. Since the study promoted the use of surface topography to visualize spinal motion during gait, the methods did not clear. Please find the suggestions below.
Abstract
1. The title could be more structured for readability.
2. Since a single paragraph of about 200 words maximum is suggested by the journal, the study should be precise and short.
3. The authors did use the style of structured abstracts, but headings like Background:, Methods:, Results:, Conclusions: should be removed as instructed by the journal.
Introduction
1. The introduction you've provided is well-written and contains important details that highlight the significance of studying segmental spinal motion during gait, particularly in relation to low back pain (LBP). However, there are a few areas that could be improved for clarity, flow, and depth. For example, the term "passenger unit" is appropriate, as it reflects the traditional view of the spine as a passive element in gait analysis. However, this concept could be expanded to highlight its limitations, especially considering current research recognizing the spine's active role in balance and locomotion dynamics.
Materials and methods
2. Rather than start the sentence with the DIERS formetric III 4D™, the DICAM v3.7 Beta analyzing system was used to collect 126 ST data from 201 asymptomatic participants. The authors should add the name of the function of the device, such as the 4D Spine & Posture Analysis device, followed by the parentheses (DIERS formetric III 4D™, DICAM v3.7 Beta analyzing system, with its company and country of origin).
3. The authors should add the subheadings of the first paragraph, such as 2.1 Equipment and experimental procedure, and explain more about the device characteristics and properties.
4. The sentence “An integrated ZebrisTM foot pressure measuring plate...” should be cut into the new paragraph.
5. The company and country of origin of the foot pressure device should be reported in parentheses after its name.
6. No information about participants’ preparations and experimental procedures is reported for both devices (spine & posture analysis and foot pressure).
7. Please add a flowchart illustrating the methods that could help the audiences understand the study.
8. Adding figures of the devices and their measures could help the audiences understand the nature of the information received from each device.
9. In section 2.2. Assembly of individual measurement export files and creation of rotation graphs, please specify which data were received and processed.
10. Although the authors used an openly available SAS (v9.4) syntax script, you should briefly describe the software purpose and how to process.
11. In the abstract, the data were reported as obtained from 201 healthy asymptomatic participants (132 females, 69 males), but this information did not show in the methods.
12. The study should report on experiments and how to get data and can add information that those data are openly to be downloaded.
Results
1. All the figures are small; they can be enlarged to make them clearer.
2. In section 3.1. Raw data visualizations of rotational curves directly related to phases gait, the authors should provide more details of data, such as the number of participants, the number of cycles, and relevant information.
3. The terms Nr. 641, Nr. 622, and so on were not clear what they referred to; please clarify.
4. In the figures shown on the standardized gait cycle, the authors should clarify the percentage of stance and swing phases.
5. Please clarify how many participants are derived from the data shown on each figure.
Discussion and conclusion
From this, “With the described methodologic advancements, ST can be used to visualize spinal motion as it directly relates to phases of gait and after standardization, to compare these results intra- and inter-individually," the statement implies a need to analyze spinal motion patterns and their variability within and between individuals. Based on this, several statistical analyses could be applied depending on the specific research questions. If no statistical analyses are performed, the study risks being perceived as incomplete or unscientific. The findings would lack the quantitative rigor required for validation, hypothesis testing, and generalization.
Reviewer 3 Report
Comments and Suggestions for Authors
- Quality of the figures: The grey background should be removed.
- Figures 2, 3, 4, and 5: These figures have nearly identical axes; only the titles differ. The authors should add additional parameters to better differentiate these figures.
- Figures 6, 7, 8, and 9: The same comment applies.
- The authors should provide more details (and include a reference) regarding the healthy subject case, as the study primarily relies on data from healthy participants. This could limit the applicability of the results to populations with pathological conditions (e.g., lower back pain or other musculoskeletal disorders).
- The study mentions that there is no alternative gold standard to validate the hypotheses made in the analysis. This limitation implies that the conclusions are based on internal validity measures rather than comparisons with an established criterion. The authors should clarify this point.
- The authors should include a comparison of their method with other approaches used for gait analysis.
- In the discussion section, the authors should frankly address the limitations of their method.
Round 2
Reviewer 1 Report
Comments and Suggestions for Authors
-
Reviewer 2 Report
Comments and Suggestions for Authors
Thank you for your thorough revisions and for addressing the comments provided in the previous review. However, the size of some figures is quite small, leading to difficulty looking at their details. In figure 5 and 9, adjusting the ylim is needed to show the whole plot.
Reviewer 3 Report
Comments and Suggestions for Authors
Thanks for the point-by-point responses.
1) For the comment 7:
Comment 7: In the discussion section, the authors should frankly address the limitations of their method.
Your answer: “Thank you for encouraging us to address more limitations! As can be seen above (replies 4b-6), we have already added many further limitations to the discussion section. Additionally, we think that we can highlight one important aspect at an even more salient position of this manuscript, that is the last sentence. It now reads: Nevertheless, before dynamic ST has been validated with a gold-standard measurement, crucial clinical decisions should not solely rely on its results.”
The answer that you give is not convincing.
2) The conclusion should include results and future work.
Round 3
Reviewer 3 Report
Comments and Suggestions for Authors
Thanks for the point-by-point responses. All comments have been responded to. The manuscript quality is enhanced based on the revised version. The paper can be accepted for publication.